# Extracellular Vesicles: New Endogenous Shuttles for miRNAs in Cancer Diagnosis and Therapy?

**DOI:** 10.3390/ijms21186486

**Published:** 2020-09-04

**Authors:** Stefano Martellucci, Nicola Salvatore Orefice, Adriano Angelucci, Amalia Luce, Michele Caraglia, Silvia Zappavigna

**Affiliations:** 1Department of Biotechnological and Applied Clinical Sciences, University of L’Aquila, 67100 L’Aquila, Italy; stefano.martellucci@uniroma1.it (S.M.); adriano.angelucci@univaq.it (A.A.); 2Department of Medicine, University of Wisconsin-Madison, Madison, WI 53705, USA; 3Waisman Center, University of Wisconsin-Madison, Madison, WI 53705, USA; 4Department of Precision Medicine, University of Campania “L. Vanvitelli”, 80138 Naples, Italy; amalia.luce@unicampania.it (A.L.); michele.caraglia@unicampania.it (M.C.); silvia.zappavigna@unicampania.it (S.Z.); 5Biogem Scarl, Institute of Genetic Research, Laboratory of Precision and Molecular Oncology, Ariano Irpino, 83031 Avellino, Italy

**Keywords:** extracellular vesicles, microvesicles, exosomes, microRNA, nucleic acids cargo, cancer, biomarkers, diagnosis, therapy, clinical studies

## Abstract

Extracellular Vesicles (EVs) represent a heterogeneous population of membranous cell-derived structures, including cargo-oriented exosomes and microvesicles. EVs are functionally associated with intercellular communication and play an essential role in multiple physiopathological conditions. Shedding of EVs is frequently increased in malignancies and their content, including proteins and nucleic acids, altered during carcinogenesis and cancer progression. EVs-mediated intercellular communication between tumor cells and between tumor and stromal cells can modulate, through cargo miRNA, the survival, progression, and drug resistance in cancer conditions. These consolidated suggestions and EVs’ stability in bodily fluids have led to extensive investigations on the potential employment of circulating EVs-derived miRNAs as tumor biomarkers and potential therapeutic vehicles. In this review, we highlight the current knowledge about circulating EVs-miRNAs in human cancer and the application limits of these tools, discussing their clinical utility and challenges in functions such as in biomarkers and instruments for diagnosis, prognosis, and therapy.

## 1. Introduction

### 1.1. General Characteristics and Nomenclature

The secretion of extracellular membranous particles is a widespread characteristic of both eukaryotic and prokaryotic cells. Extracellular Vesicles (EVs) are small membrane-bound vesicles that are secreted from numerous animal cell types, including a variety of tumor cells [1,2,3,4,5,6]. Several EVs subtypes have been proposed with names such as ectosomes, microvesicles, microparticles, exosomes, oncosomes, apoptotic bodies, and more. To date, the International Society for Extracellular Vesicles established that subtypes of EVs may be defined by physical characteristics such as size (“small EVs”, sEVs, diameter < 100/150 nm and “medium/large EVs”, m/lEVs, diameter > 200 nm), density (low, middle, high, with each range defined), biochemical composition (CD63+/CD81 EV-associated, Annexin A5-EVs), and description of conditions or cell of origin (podocyte EVs, hypoxic EVs, large oncosomes, apoptotic bodies) [7].

In general, the common nomenclature used in literature for EVs comprises exosomes (small EVs) and microvesicles (large EVs). Exosomes are released on multi-vesicular bodies (MVBs) fusion with the plasma membrane, they are ~30–150 nm in diameter and they have “cup-shaped” morphology when observed under transmission electron microscopy. Exosome biogenesis begins during the early-endosome maturation process. Early-endosomal membranes invaginate to create intraluminal vesicles (ILVs), thereby forming MVBs. The majority of MVBs are degraded by fusion with lysosomes; however, some populations of MVBs migrate towards and fuse with the plasma membrane, releasing their ILVs (now called exosomes) into the extracellular space [7,8,9]. Microvesicle biogenesis occurs via the direct outward blebbing and pinching of the plasma membrane releasing the nascent microvesicle into the extracellular space. Microvesicles are large EVs and very heterogeneous in size (~400–1000 nm) [7,10]. Consolidated data have largely demonstrated that EVs represent an important paracrine and endocrine mechanism for cell-to-cell communication, involved in the transfer of several different macromolecules including microRNAs (miRNAs), short non-coding RNAs with key roles in cellular regulation [11,12,13,14,15].

### 1.2. Functions in Tumoral Cells

Several mechanisms have been proposed to describe possible EVs/target cell interactions:EVs could interact directly with target cells by employing conventional ligand–receptor interactions.The membrane proteins of the EVs can be cut by specific proteases giving rise to fragments that could act as soluble ligands for membrane receptors on the target cells.EVs can merge with the target cell membrane by transferring cargo proteins and RNAs [16].

In particular, miRNAs transported within EVs can regulate the expression of protein-coding genes in a paracrine way by binding mRNAs in the target cells [17]. Accumulating evidence suggests that EVs play an important role in communication between tumors and the microenvironment; indeed, by transferring EVs cargo, tumor cells can alter the function of both local and distant normal cells, thereby promoting tumor progression, immune evasion, angiogenesis, and metastasis [14,18,19,20,21].

Due to the stability of EVs miRNAs in bodily fluids, and their functional association with cancer, circulating EVs miRNAs are now extensively investigated for their potential use as cancer biomarkers in diagnostics and several EVs isolation methods have been developed for the purification of exosomes and microvesicles from bodily fluids and/or cell culture media. Indeed, to date, several miRNAs contained within EVs have been consistently associated with cancer progression, metastasis, and aggressive tumor phenotypes [22,23,24,25,26,27].

The role of signal transducer renders EVs cargo a diagnostic tool and a fertile field of study in innovative therapy. Indeed, great interest has arisen in the possible clinical use of EVs in regenerative medicine thanks to their ability to act as paracrine mediators in cell–cell communication [28,29,30].

The release of EVs by cancer cells could play an important role also in protecting tumor cells from apoptosis and making them more resistant to chemotherapy by extruding apoptosis-inducing proteins and chemotherapy agents [31]. In such a background, EVs gained great attention as carriers for miRNA delivery to regulate target gene expression and improve cancer therapeutic strategies [32,33].

### 1.3. Profiling EVs-miRNAs

Numerous studies have been carried out to analyze miRNAs’ involvement in human diseases through the development of a microarray platform, which made possible the global profiling of miRNAs. Significantly, in many cases, the classification of tumor phenotypes was more successful when the expression profile of miRNAs was used, rather than that of mRNAs. Although analysis of the miRNA profile in human tissues has been shown to have great potential as a disease marker, a less invasive method for tissue biopsy is certainly needed to make this analysis more affordable. In this regard, the identification of miRNA in body fluids attracted much interest. It prompted researchers in demonstrating that the miRNA profiles in the body fluids from patients were significantly different from the profiles in healthy subjects [34,35,36].

Moreover, these studies demonstrated that miRNAs in body fluids remained stable under various extreme conditions, such as boiling, a very low or high pH, repeated freeze–thaw, and storage at room temperature for a long time. Besides, thanks to the high sensitivity and speed of the new profiling techniques of molecular biology, today it is much easier to study the profile of the global miRNA level, rather than that of proteins or secreted metabolites [37,38].

Careful consideration of the EVs isolation method must be taken into account when interpreting study results. The most common methods are a series of differential centrifugation steps, ultrafiltration, chemical affinity purification, and Exoquick [39,40,41] (Table 1). As demonstrated in conditional medium from cell culture the purification methods of EVs may heavily influence the yield, purity, and integrity of RNA extracted from EVs [42,43]. Moreover, there are no reliable methods for purifying and discriminating between exosomes or microvesicles, and this could represent an important limitation in many studies. Indeed, even though both exosomes and microvesicles have been thought to share intercellular communicative ability and they present largely similar RNA profiles, a recent study points out that exosomes and microvesicles could have different biological roles. The study, which was performed using transiently transfected cells, showed that reported proteins and mRNA were successfully sealed in both types of vesicles but only microvesicles managed to convey reporter function to target cells [44,45,46].

To date, an elevated number of miRNAs have been associated with tissue expression patterns and linked to specific pathologies. Thus, their analysis can provide a large amount of information in a relatively short time. Tumor-related alterations in microRNA expression and function can reflect molecular processes of tumor onset and progression qualifying microRNAs as potential diagnostic and prognostic biomarkers. Current investigations show that miRNAs are detectable in different tissue types and a wide range of biological fluids, either free or trapped in circulating EVs [47,48].

Their abundance in circulating body fluids in conjunction with the feasibility in the recovery and characterization from liquid specimens render EVs cargo molecules promising diagnostic and prognostic biomarkers for cancers. The traditional enrichment and extraction processes of miRNA trapped inside EVs, are problematic due to the low concentration of miRNA within EVs and the lack of a standardized isolation method of EVs. Indeed, some authors, considering these difficulties, are skeptical about the possibility to translate EVs-miRNA analysis in the clinic. This is also justified by several unsolved problems, such as significant differences in the procedures for processing samples, methods of analysis, and high heterogeneity in results accumulated so far [49]. To solve the aforementioned problems, current studies are mainly focusing the attention on the needed improvements in methodology able to increase the reliability of EVs-miRNAs as a diagnostic tool [49]. A detailed critical discussion on methodology challenges is outside the purpose of this review and further details could be found in several studies cited in the text while advice about these aspects is present in MISEV2014 guidelines [7].

In biological samples, such as liquid biopsies, EVs-miRNAs can be analyzed through different techniques. Evaluation of plasma EVs-RNA profiles by RNA-seq individuated a variety of RNA species for their abundance, stability, and age/gender correlation as well as disease association. However, the experimental procedures can cause constitutive variations in EVs-miRNA profiles if using different methods for RNA extraction, sequencing library preparation, or gel size selection [50].

Several traditional methods have been employed for miRNA detection, including Northern blotting [51,52,53], microarray analysis [54] and quantitative polymerase chain reaction (qPCR) [55,56].

So far, qPCR is regarded as the gold standard. However, it has been associated with many limitations such as its time-consuming nature, its false-positive propensity, and difficulties in primer design. qPCR presents the main advantage of being highly sensitive and specific. The newly introduced next-generation sequencing (NGS) has recently proven to be extremely able to detect unknown miRNAs that the traditional methods, including the qPCR, are incapable of doing. miRNAs detection by NGS has progressed rapidly and is a promising field for applications in drug development [57,58] (Table 1).

Recent studies that have investigated miRNA profiling by NGS, with particular attention to the potential applicability on biofluids, aimed at comparing results with those obtained by qRT-PCR [59].

The high sensitivity of new molecular techniques suggests the adoption of accurate pre-analytical processing of EVs. Indeed, it has been suggested that EVs may associate with protein complexes and lipoproteins bound to free circulating miRNAs, suggesting the use of proteinase K and RNase A before the analysis of miRNA-EVs to remove the extra-vesicular contamination source. Indeed, the proteinase and RNase treatment can reduce the relative quantity value by up to 70% for different miRNAs [60].

The utilization of new technologies producing a huge amount of data generates new problems concerning statistical analysis and data interpretation. It could be a difficult procedure to select the miRNA profile that effectively plays a role in tumor development and progression, without incurring bias. A single miRNA can target many mRNAs and modulate many proteins involved in different cancer-related pathways. Moreover, some miRNAs can act as both oncogenes and oncosuppressors, according to the biological context, and this makes it more complicated to unveil their effective role. To identify potential biological functions and potential targets of miRNAs several software programs such as TargetScan, miRbase, miRWalk, DIANA-microT, and TarBase have been described, and for more information about this software, we suggest the reading of specific publications [28,29,61,62].

## 2. EVs-miRNAs as Biomarkers in Bodily Fluids

### 2.1. EVs-miRNAs in Blood

Liquid biopsy is a new promising diagnostic approach in oncology including the analysis of biomarkers shed from primary or metastatic tumors into the peripheral blood. The expression profiles of blood miRNAs have been shown to vary widely between normal and tumor patients and thus many studies have investigated whole blood as an accessible source of cancer fingerprint [63].

Both plasma and serum have been analyzed equally in the published studies, also in parallel, generating results that are not always comparable. In recent years, plasma was preferred to serum as a source for EVs to avoid contamination with coagulation-associated EVs released from platelets or with lysed blood cells.

Effective identification of miRNAs in circulating EVs was described in several tumors, including lung cancer, renal cancer, and ovarian cancer [64,65,66]. Taylor et al. [66] have observed, for example, that the expression profile of miRNA in EVs released into the circulation by ovarian cancer reflects the profile of the tumor itself, suggesting their possible use in screening for the detection of ovarian cancer in asymptomatic individuals and to monitor the risk of recurrence.

#### 2.1.1. Reliability and Feasibility Issues

Many of the current efforts are being made for improving the reliability and feasibility of the methods, a necessary step in a long way for clinic translation. For example, in prostate cancer, the role of “prostasomes”, the prostate EVs cargo, is currently being investigated as a diagnostic tool in two clinical trials. Both of them have been designed to recruit patients with prostate cancer and analyze the prostasome content by NGS (https://clinicaltrials.gov Identifier: NCT03694483, NCT03911999). However, many aspects of the effective role of EVs-miRNA as blood-derived cancer biomarkers should be clarified. Currently, it is still debated if the EVs-based miRNA detection assays are superior to the whole serum/plasma-based assays [67].

A central question is about the concentration of miRNAs, that have been calculated also far less than one molecule per EVs [68]. This aspect is particularly unfavorable in a diagnostic procedure when associated with the high variation in EVs yield and isolation method and it may compromise the reproducibility of the results. Besides, the analysis of putative tumor tissue-derived EVs-miRNA could receive a significant interference by miRNA from hemolysis in cancer patients but also by other non-tumor cells [69,70]. An ideal solution for this problem could derive from the utilization of reference miRNAs. However, it is difficult to define an optimal endogenous gene for normalizing exosomal miRNA expression. According to many reports, miR-16-5p, miR-423-3p, and miR-191-5p are constantly expressed and usually used as reference miRNAs. On the contrary, snRNU6 is not adequately expressed in most blood samples. Since these reference genes are not always adequate for all samples but vary according to populations and their characteristics, it is necessary to validate them for each dataset. It is also possible to use some dedicated software such as the Genome Algorithm to select the most suitable miRNAs [42].

Furthermore, researchers often use an exogenous normalizer to eliminate or reduce the technical noise and inter-individual variability, e.g., cel-miR-39.

#### 2.1.2. Diagnostic Power

Different studies have highlighted the many diagnostic advantages offered by the analysis of nucleic acids within EVs with respect to free-circulating forms:EVs content better resembles the biology of the cells it originated from.EVs represent a protected environment for nucleic acids, avoiding modification.Profiling EVs cargo can permit the identification of the molecular signature of its tissue source [71,72]. Comparative studies have frequently demonstrated that miRNAs detected in EVs enriched specimens are more representative with respect to whole serum. ElSharawy et al. compared miRNA signature in whole serum, particle concentration, and particle depleted fractions from ten patients with colorectal cancer (CRC), and they were able to identify about twofold more miRNAs in the particle concentrated fraction with respect to the number of miRNAs in whole sera. These authors individuated a large set of differentially expressed miRNAs only in the patient’s particle-concentrated sera, confirming the value of candidate markers such as miR-15b, miR-21, miR-25, miR-92, miR-93, miR-223, and miR-486. Besides, they found that more than two-thirds (15/22) of these miRNAs were consistent in CRC-patients’ particle-concentrated sera and the matched tumor tissue samples [73].

Additionally, Cheng et al. demonstrated that miRNA levels differ remarkably between plasma and serum EVs and between EVs and cell-free plasma/serum from three healthy control patients [74]. Overall, the number of miRNAs identified in this study was 943 and EVs isolated from serum contained the highest percentage of miRNA (>40%) compared to plasma-EVs (30%) and free circulating serum/plasma miRNA (13–14%). According to certain dimensions (70–110 nm), EVs analyzed in the study by Cheng et al. could be considered exosomes. The most abundant miRNAs found in all samples were hsa-miR-451a, hsa-miR-191-5p, hsa-miR- 486-5p, hsa-miR-223-3p, and hsa-miR-484. Pathway prediction according to Encyclopedia of Genes and Genome (KEGG) demonstrated that unique exosome-miRNA targeted neuronal signaling through GABAergic, axon guidance, endocytosis, and glutamatergic synapse signaling pathways.

Endzelins et al. obtained opposite results. They isolated RNA from whole plasma and plasma EVs samples from patients with prostate cancer and benign prostatic hypertrophy patients and performed an RT-qPCR analysis of nine selected miRNA biomarkers. To eliminate RNA contamination, they performed a proteinase and RNase pre-treatment that reduced up to 70% of relative quantity value for different miRNAs associated with EVs. They found that the EVs-miRNA profile differed significantly from the cell-free miRNA profile in the whole plasma; EVs-miRNA represented only a small fraction of the cell-free miRNA [60].

Thus, deep efforts must be proposed to validate EVs-miRNAs from the liquid biopsy. For example, some studies reported that miRNAs extracted from exosomes did not correspond to the transcripts in primary tumors [75]. Bovy et al. found that exosomal miR-503 levels increased in patients after neoadjuvant chemotherapy, while the expression of miRNA in primary tumors was not modified. The authors hypothesized that circulating miR503 did not derive from primary tumors but endothelial cells. Several cell types may secrete exosomes, and we should improve our knowledge about the mechanisms of miRNA sorting into EVs to understand their role in tumor progression [76]. To overcome this issue, many studies tried to identify surface markers on exosomes to distinguish cancer-specific exosomes from other EVs but no promising results have been achieved [77,78].

#### 2.1.3. Validation Strategies

Different approaches for validating EVs-miRNA profiles have been proposed, including the comparison with profiles from tumor cell lines and/or primary tumors, or with the published databases. These studies have frequently followed complex and non-homogeneous protocols and thus their results are not easily comparable. Ogata-Kawata et al. first compared diagnostic miRNA profiles between serum and culture medium from CRC patients and colon cancer cell lines, and then verified the 16 common up-regulated miRNAs with EVs-miRNAs from sera of 29 pairs of patients before and after surgical resection of the tumor [79]. In this study, the modulation of miRNAs was used as a verification step of the diagnostic value of the miRNA and indicated that only seven miRNAs (let-7a, miR-1229, miR- 1246, miR-150, miR-21, miR-223, and miR-23a) were significantly higher in serum exosomes of primary CRC patients, compared to healthy controls. Chen et al. followed a different validation strategy, starting their analysis from the in vitro approach. Indeed, they performed a correlation analysis of miRNA from two CRC lines with CRC tumor miRNA profiles obtained from The Cancer Genome Atlas (TCGA) representing normal colon tissues and different CRC stage tumors [44]. They found that 50 of the EVs-enriched miRNAs were dysregulated in different CRC stages compared to normal colon tissue. However, the difficulty in the interpretation of results from different experimental approaches became apparent when Chen et al. compared the 97 EVs-enriched miRNAs from CRC lines with EVs-miRNA profiles of CRC patients reported in previous studies. In comparison with the plasma EVs-miRNA signature reported by Ogata-Kawata, Chen demonstrated the presence of only four miRNAs commonly enriched in EVs (miR-150-5p, miR-1246, miR-766-3p, and miR-10b-5p) [79]. Moreover, the comparison with the serum EV-miRNA signature reported by Yuan et al. considering also different stages CRC patients, demonstrated that miR-1246 was the only abundant EVs-miRNAs in all CRC line subtypes and up-regulated in stage II, III, and IV CRC patients. Besides, several EVs-enriched miRNAs were found to be up-regulated only in advanced CRC patients, such as let-7b-3p, miR-27a-5p, miR-182-5p, miR-192-5p and miR-486-5p [44,50]. This latter observation confirms the low predictivity of tumor cell lines, probably due to a restricted phenotype that in the majority of the cases resembles only the advanced heterogeneous stages of tumors.

#### 2.1.4. Early Diagnosis

EVs-miRNAs have been investigated for their role of biomarkers for both the early stages and prognosis of advanced cancer. miR-375 was significantly increased in prostate cancer patients compared to benign prostatic hypertrophy only when tested in whole plasma, on the contrary, miR-200c-3p and miR-21-5p could differentiate between prostate cancer and benign prostatic hypertrophy better when tested in EVs than in the whole plasma. Besides, they found that the level of Let-7a-5p was significantly decreased in EVs from prostate cancer patients with high Gleason score (≥8) compared to low Gleason score (≤6). These contradictory results suggest that the pre-analytic processing of the serum/plasma is an important issue in reproducibility, affecting also the recovery efficiency of EVs-miRNAs [60].

It was proposed that EVs-miRNAs could represent an effective tool for individuating latent stages during carcinogenesis, increasing our diagnostic power. For example, in 2017, Jin et al. selected four exosome-derived miRNAs (let-7b-5p, let-7e-5p, miR-23a-3p, and miR-486-5p) that were promising for early diagnosis of non-small cell lung cancer (NSCLC). The profile of all identified microRNAs showed sensitivity and specificity higher than 80% in NSCLC identification [80]. In addition, real-time RT-PCR analysis of EVs-enriched plasma miRNA in 50 early-stage CRC patients (stage I and II) vs. 50 healthy subjects revealed that miR-125a-3p and miR-320c were significantly up-regulated in cancer patients. The diagnostic power for CRC cancer determined by ROC curve was 68.5% AUC for miR-125a-3p and 60.0% for mir-320c, lower than that of reference biomarker CEA (AUC = 83.6%). Finally, in contrast to CEA, miR-125a-3p and miR-320c did not show a significant correlation with tumor size and differentiation degree [81]. Li et al. found that miR-20b-5p and miR106a-5p were overexpressed in both plasma and serum from breast cancer patients compared to healthy subjects, and diagnostic performance of the combination using AUC analysis was 83–88% (plasma) and 92–96% for serum. The same authors demonstrated that both miR-20b-5p and miR106a-5p were overexpressed in EVs-enriched sera from breast cancer patients respect healthy donors [82]. In glioblastoma, one important study by Manterola et al. described that the combination of two EVs-miRNAs (miR-320 and miR574-3p) and RNU6-1 may discriminate glioblastoma patients from healthy individuals [83].

#### 2.1.5. Cancer Staging

Moreover, the analysis of the EVs-miRNA signature could help identify cancer staging and prognosis [84]. For example, Kanaoka et al. used a retrospective analysis to identify EVs-miR-451a as a promising biomarker to predict NSCLC patients’ prognosis at stages I, II, and III [85]. In breast cancer, exo-miR-223-3p is associated with histological type pT and pN, pathological stages of lymphatic invasion, and the nuclear grade of breast cancer, resulting in a significantly higher early stage [86]. In prostate cancer, miR-182 and miR-183 are highly expressed in cancer exosomes [87], while miR-1290 and miR-375 may predict prognosis in patients with prostate cancer [88]. Indeed, EVs-miRNAs can act as mediators for intercellular communication promoting metastatic progression. When EVs-miR222, overexpressed in multiple types of cancer, was analyzed in plasma from breast cancer patients with and without lymph node metastasis, the highest level of miR-222 was metastatic patients. In the same study, the authors confirmed the EVs-miR-222 overexpression in more invasive breast cancer cell lines [89].

Surgical resection of glioblastoma was correlated to down-modulation of serum EVs-miR-301a, and the increased expression of the same miRNA was associated with the recurrence of the malignancy. For this reason, EVs-miR-301a was proposed as both a diagnosis and recurrence biomarker in glioblastoma [90]. Meng et al. found that it was possible to discriminate advanced stage III-IV ovarian cancers from early stages cancers measuring exosomal miR-200b and miR-200c [91]. Other microRNAs have also been selected for their potential use as diagnostic serum biomarkers useful in discriminating different cancer subtypes. For example, in breast cancer, exo-miR-373 was useful in individuating triple-negative breast cancer (TNBC) subtypes with respect to luminal subtypes [92,93]. In NSCLC, Jin et al. identified four exo-miRNAs (miR-181-5p, miR-30a-3p, miR-30e-3p, and miR-361-5p) and three exo-miRNAs (miR-10b-5p, miR-15b-5p, and miR-320b) that could be used as specific signature of adenocarcinoma and squamous cell carcinoma, respectively [80].

#### 2.1.6. Predictive Ability

Specific EVs-miRNAs have been identified as recurrence-specific or predictive biomarkers. The use of EVs-miRNAs as predictive biomarkers, when combined with innovative biotechnological analytical methods, assure a revolution in the field of oncological therapy, guiding the best curative protocol. In NSCLC patients, miR-4257 and miR-21 are more expressed in patients undergoing recurrence after surgery compared to patients without recurrence [94]. Svedman et al. using a polymeric precipitation technique obtained EVs-enriched plasma samples from metastatic cutaneous malignant melanoma collected before and during MAPK inhibitors therapy. In their study, Svedman et al. found that increased levels of EVs-miRNAs let-7g-5p and miR-497-5p were associated with better disease control [95]. Inhibition of BRAFV660E with vemurafenib in melanoma cells determined an increased secretion of EVs, and in an important manner, also significant changes in the RNA cargo of EVs. This phenomenon might lead to altered gene expression in the cancer population that reduces sensitivity to the inhibitor. Lunavat et al. showed that miR-211–5p expression, detectable in EVs-enriched in vitro and in vivo specimens, was induced upon BRAFV600E inhibition due to the up-regulation of the master regulator MITF, promoting survival and proliferation of melanoma cells [96]. In 2018, at Lei Li, Peking Union Medical College Hospital started a prospective enrolment of high-grade serous ovarian cancer (HGSOC) and benign gynecologic disease patients to identify, with the aid of next-generation sequencing (NGS), miRNAs and ncRNAs that were biomarkers for the detection and prediction of progression-free survival of ovarian cancer (https://clinicaltrials.gov Identifier: NCT03738319). In addition, Kuhlmann et al. referred to an NGS-based workflow for analyzing the signature of EVs-associated miRNAs in the plasma of platinum-resistant ovarian cancer patients. After a comparison of different EVs-enrichment methods, authors chose Exoquick reagent, and they observed a specific miRNA profile (miR-181a, miR-1908, miR-21, miR-486, and miR-223), which was differentially abundant in the plasma of platinum-resistant patients [97].

### 2.2. EV-miRNAs in the Urinary Specimen

miRNAs collected from urinary specimens can provide valuable evidence concerning tumors from the urinary system, due to their many consolidated advantages including long-term stability and noninvasiveness. This would be helpful not only for diagnostic and follow-up purposes but also for therapeutic decisions, as miRNAs also have been linked to clinical outcomes in urothelial carcinoma [98]. The underlying rationale is that a single specific miRNA or a signature of multiple miRNAs may improve the risk stratification of patients and may supplement the histological diagnosis of urological tumors [99,100]. For example, the identification of biomarkers characterizing the invasive potential of bladder cancer based upon identification of miRNAs from tissue samples as well as from urinary EVs are promising biomarkers and could support decision making for or against aggressive therapies.

#### 2.2.1. Bladder Cancer

Baumgart et al. defined a miRNA panel in tumor tissues as well as in urinary EVs for discriminating muscle-invasive bladder cancer from non-muscle-invasive bladder cancer [101]. For the detection of bladder cancer in urine, voided urine cytology serves as the gold standard. Despite its high specificity, voided urine cytology has shortcomings in terms of sensitivity. So, Erdmann et al. evaluated the diagnostic potential of the urinary levels of selected miRNAs, finding that miR-125b, -145, -183, and -221 in combination with voided urine cytology could have a great diagnostic potential for non-invasive detection of bladder cancer in urine [102].

Detection and quantification of cancer miRNAs in urine can be performed using the cellular sediment, which also contains cells, or EVs originating from those cells [103].

When miRNAs have been investigated as possible biomarkers for bladder cancer in both urine and plasma, results were only partially overlapping. Sabo et al. evaluated the expression levels of miRNAs contained in plasma EVs from 47 men with bladder cancer and 46 healthy controls by next-generation sequencing. The miRNA profiles were compared with urinary profiles from the same subjects. Potential plasma EV-associated biomarkers such as miR-3140-3p and miR-454-5p were not detected in urine samples while miR-628-3p and miR-4508 were significantly upregulated in urine, in contrast to plasma where they were downregulated. Only miR-126-3p was significantly upregulated in bladder cancers with respect to controls in both plasma EVs and urine. Although in this study authors did not specifically evaluate urinary EVs-miRNAs, but urinary cell-free miRNAs, results suggest caution in translating plasma results to urinary miRNA profiles [104]. Similar results were obtained in type 2 diabetes mellitus by Park et al. that first compared various isolation methods and then evaluated the correlations of EVs miRNA between urine and serum samples. They found that the ultracentrifugation method yielded the highest number of EVs-miRNA molecules and that EVs-miRNA profiles showed high variability in correlation coefficients between urine and serum EVs-miRNAs for each patient [105].

Xu et al. presented a new method to collect low concentrations of miRNAs from dilute solutions such as urine. Magnetic nanoparticles with a 10-nm core size with carboxylic acid coating can adsorb low-concentration proteins, and form a coating formed by proteins bonded to the surface of nanoparticles (in biological fluids, also known as protein corona) which makes them easy to aggregate and precipitate for subsequent isolation. In urine, these nanoparticles can aggregate with proteins, including miRNAs-associated protein Argonaute 2 and microvesicle-related proteins, to form precipitates, so that miRNAs can be easily extracted from pellets by a small amount of lysis buffer for subsequent analysis such as real-time PCR. In the author’s opinion, this method provides an easy way to enrich miRNAs from biofluids without the need of ultracentrifugation and immunoprecipitations, bringing remarkable convenience to miRNAs-based biomarker research. [106].

#### 2.2.2. Kidney Cancer

As far as urinary exosomes are concerned, some studies have revealed a possible role of the miRNA cargo in the diagnostic and prognostic field for kidney pathologies. It has also been observed that miRNAs are involved in the development and functional maintenance of the kidney and the study of their tissue expression has made it possible to identify some profiles involved in different kidney diseases such as, for example, polycystic kidney, diabetic nephropathy, clear cell carcinoma and chromophobe carcinoma; these expression profiles could, therefore, represent useful diagnostic tools [59]. In particular, a recent study has shown for the first time the strong enrichment, in urinary exosomes of microalbuminuric patients compared to healthy control subjects, of miR-145, a mesangial cell marker whose expression is induced by TGF-β1 [107].

#### 2.2.3. Prostate Cancer

Fredsøe et al. identified and validated several deregulated miRNAs in urine samples from prostate cancer patients, profiling the expression levels of 92 miRNAs. Additionally, the authors trained both a novel diagnostic three-miRNA model (miR-222-3p*miR-24-3p/miR-30c-5p) that distinguished benign prostatic hyperplasia and prostate cancer patients and a novel prognostic three-miRNA model (miR-125b-5p*let-7a-5p/miR-151-5p) that predicted time to biochemical recurrence after radical prostatectomy [108].

### 2.3. EV-miRNAs in Other Bodily Fluids

Although blood (serum or plasma) and urine are the most common biological matrixes on which scientists focused their studies so far, other accessible liquid sources are potentially available for miRNAs detection, including cerebrospinal fluid or seminal fluid [109]. Interestingly, the sensitivity and specificity of biomarkers from the cerebrospinal fluid of patients with brain tumors is typically higher than those detected in the peripheral blood and other biological sources [110].

#### 2.3.1. Seminal Fluid

Seth et al. analyzed human seminal fluid, enriched for exocrine and other constituents of the prostate, such as prostate cancer cells, proteins, and metabolites, demonstrating the potential of seminal fluid miRNAs as diagnostic biomarkers of prostate cancer. Specifically, seminal fluid-derived miR-200b, miR-200c, miR-30a, miR-375, and miR-99a were found at higher levels in men with elevated PSA levels and biopsy-proven cancer compared with men with elevated PSA levels but no cancer [111]. Although there are not specific studies so far, it is plausible that together with cell components, EVs are also present in the seminal fluid as a consequence of the increased shedding activity of cancer cells.

#### 2.3.2. Pleural Lavage

Roman-Canal et al. developed a novel approach to identify highly sensitive and specific biomarkers by investigating the use of EVs, isolated from the pleural lavage, as a source of potential biomarkers. The differential expression analysis conducted by the authors yielded a list of 14 miRNAs that were significantly dysregulated and, between them, miRNA-1-3p, miRNA-144-5p and miRNA-150-5p were found to be the best in terms of accuracy as lung cancer diagnostic biomarkers [112].

#### 2.3.3. Nipple and Lacrimal Fluids

The ductal lavage and nipple aspirate fluids represent excellent sources for miRNA detection because they can be obtained using poorly invasive techniques and, also, they bypass the limitations of blood biomarkers evaluation with regard to the specificity of the original tissue. Indeed, Do Canto et al. showed that breast cancer miRNA screening is feasible by analyzing miRNAs in the breast ductal fluid obtained by ductal lavage. The authors analyzed hundreds of miRNAs expression in the ductal lavage fluid collected from women with breast cancer [113].

In addition, to explain the differences in clinical behavior between adenoid cystic carcinoma of the salivary gland, lacrimal gland, and breast, Andreasen et al. characterized the global miRNA expression profile in these types of cancer. The authors showed that miRNA expression profiling distinguishes these tumors based on the site of origin and that differentially expressed miRNAs are involved in the regulation of cellular processes that could be responsible for a more aggressive clinical course in tumor relative to those with better prognosis [114].

The main studies about EV-associated miRNAs as diagnostic or prognostic biomarkers from different bodily fluids are summarized in Table 2.

## 3. miRNA-Based Cancer Therapies

Cancer progression is supported also by aberrant miRNA profiles, so, therapeutic intervention could be intended to reestablish normal miRNA levels [115,116].

miRNAs target specific molecular pathways by acting both as oncogenes or tumor suppressors, therefore, to restore physiological miRNA levels in tumors two opposite strategies are used either by replacing or inhibiting miRNA activity [117,118,119,120].

The miRNAs that are upregulated in cancer act as oncogenes and are consequently named as oncomiRs [118].

They induce tumor proliferation by decreasing tumor suppressive gene levels. Therefore, specific miRNA antagonists, such as anti-miRNAs, locked-nucleic acids (LNA), or antagomiRs are delivered to inhibit oncomiRs’ expression [121,122].

miRNAs that function as tumor suppressors inhibit cancer proliferation by targeting cellular oncogenes. Their expression is lower in cancer than in normal adjacent tissues; thus, a replacement strategy that restores normal miRNA levels may represent a beneficial therapeutic tool [123].

Exogenous therapeutic miRNAs are supposed to mimic the biological functions of endogenous counterparts, reducing as far as possible the risk of off-target effects. On the other hand, the overexpression of exogenously administered miRNAs might also target previously unidentified genes [124]. Besides, exogenous miRNAs might also saturate endogenous miRNA processing enzymes, probably leading to perturbation of miRNA functions [119,125].

Furthermore, circulating miRNAs may play a role in tumor development as ligands of toll-like receptors [126].

Alongside these promising characteristics of miRNAs in cancer therapy, their effective utilization in the clinic must overcome several hurdles. First, in most studies, the route of administration of miRNA-mimic and the antagomiRs is intravenous infusion. While ensuring a systemic distribution, the highest levels are obtained in the organs of clearance (kidney and liver). We, therefore, need technologies that guarantee selective distribution to the target cells, for example conjugating these molecules with peptides that determine a selective uptake by selected cells or through local administration. Secondly, important drawbacks tare associated with poor miRNA stability in vivo, modification of endogenous RNA machinery, and unnecessary side effects. The development of cancer is associated more frequently with the altered expression of multiple miRNAs. The interaction between the different miRNAs should, therefore, be considered before acting on the individual miRNA. However, it is now clear that to achieve successful results the development of an effective and safe delivery system is needed, including the utilization of EVs as vectors [127].

### 3.1. Strategies for RNA Loading into EVs

Below 10% of EV-RNAs relates to the whole parental cell transcripts, suggesting that a cellular program for loading RNA into EVs exists [128,129,130]. Such a program, as demonstrated also by studies about DNA packaging into EVs, permits the loading of sequences less than 700 nucleotides long [131,132], such as miRNAs and siRNAs, protecting them by endonuclease degradation [133] but is not suitable to deliver full coding sequences. However, these selective mechanisms responsible for RNA loading into EVs are under investigation, and it is currently known that a specific GGAC sequence, called EXOmotif, drives the packaging of miRNAs into EVs [134,135].

The accurate quantification of RNAs loaded in EVs is challenging and further complicated by technical frailties and the lack of specific standards. However, it has been demonstrated by analyzing a purified EV population that less than one molecule of a given miRNA was present per EV [68] suggesting either the existence of a subpopulation of vesicles containing significant amounts of miRNAs or the loading of only a few miRNA molecules in a single EV [136,137]. At any rate, this quantity of RNA in EVs is insufficient to effectively modulate gene expression [138], but if we consider that up to 500 copies of miRNAs may be loaded into a single vesicle, up to 30,000 EVs per day may be produced by a single cell, and that in some cases cancer microenvironment stimulates significantly the release of EVs, the probability of an effective RNA transfer [130,139] significantly increases.

On the aforementioned basis, it was suggested that the miRNA amount loaded into EVs should be increased by re-engineering naturally-derived EVs to ensure a sufficient miRNA payload for clinical anti-tumor therapies [75,140,141]. To this end, two main methods for uploading miRNAs into EVs were established: the first, known as preloading or the endogenous method, requires that the EVs-producing cells are genetically modified; in the second, called post-loading or exogenous method, purified EVs are loaded with miRNAs [142,143].

Both the strategies have advantages and limitations; therefore, they are selected based on the type of cells to be used for EVs purification and/or the type of the genetic material to deliver. For example, the pre-loading approach allows us to generate genetically modified cell lines that produce EVs by using stable transfection or viral-mediated gene transfer since it is challenging to safely and efficiently transfect primary cells, the post-loading approach can be more appropriate.

#### 3.1.1. Pre-Loading Approach

The pre-loading method is based on the findings that miRNA overexpression into cells induces the increase in its content in EVs [144]. Genetic modification of EV-producing cells by transfection with miRNA-expressing plasmids or viral vectors or by inserting exogenous miRNAs increases miRNA levels that are incorporated into secreted EVs. The potential influence of transfection reagents on EVs packaging should be taken into consideration [145]. Cellular processes of miRNAs sorting into EVs are poorly characterized; thus, it is not known if certain families of miRNAs may be preferentially packed into EVs, rendering this process only appropriate for certain groups of miRNAs [83,146]. It should also be remembered that most miRNAs are not associated with EVs in the cell supernatant; thus, it is complicated to efficiently load miRNAs into EVs. Moreover, the pre-loading approach requires massive optimization in each different cell type suitable for producing EVs [1].

#### 3.1.2. Post-Loading Approach

The post-loading approach involves the introduction of synthetic miRNA oligonucleotides by transfection or electroporation into previously isolated EVs. This method should represent a more controlled procedure than the pre-loading approach, even if several issues concerning EVs and miRNAs may emerge [147]. During the loading of siRNAs into EVs, the electroporation buffer may induce RNA precipitation and aggregation by decreasing the loading efficiency below 0.05% [148]. Optimization of the experimental conditions allowed us to reach a 55% loading efficiency into EVs [149]. On the other hand, several studies have demonstrated that electroporation was not suitable for the efficient loading of miRNA into EVs and selected transfection to genetically modified cells [150]. Recently, a transfection method based on a calcium phosphate co-precipitation plus heat shock approach for uploading miRNAs into purified EVs has been described [151]. However, caution needs to be exercised in figuring out the loading efficiency due to the feasible presence of complexes among transfection reagents and miRNAs not enclosed into EVs. The disagreement in the miRNA loading efficiency observed by different groups may be further attributed to variations in EVs-producing cells and/or in EVs purification methods.

### 3.2. EVs for Therapeutic microRNA Delivery

Preclinical studies based on the use of EVs for the delivery of nucleic acids have shown promising results for the treatment of tumors [75,152]. Here, we will describe the most recent studies in which miRNAs or antagomiRs have been incorporated into EVs and used as therapeutic agents.

One example is represented by EVs naturally released by adult liver stem cells, capable of inhibiting the proliferation of hepatoma cells both in vitro and in vivo by delivering specific miRNAs [153]. Besides, in vitro studies showed that miRNAs incorporated into EVs and transferred to target cells, such as breast cancer or osteosarcoma cells, were able to specifically modulate gene expression. [141,154].

In addition, in several studies, treatment with EVs incorporating specific miRNAs was able to reduce cell proliferation in different tumor models. For example, EVs isolated from adipose tissue-derived stromal cell medium after transfection with a plasmid expressing miRNA-122 induced sensitization of hepatocellular carcinoma cells to Sorafenib both in vitro and in vivo [155,156].

EVs purified from bone marrow-derived stromal cells and enriched in miR-146 by intra-tumor injection inhibited the growth of a primary brain tumor in vivo in a rat model [157].

Furthermore, EVs have been modified to ensure specific targeting of some tumor models, for example by favoring the delivery of miRNAs in breast cancer cells [158,159,160] (Table 3).

Besides, the use of EVs incorporating antagomiRs that downregulated miR9 and miR150 levels provided promising results in models of glioblastoma and sarcoma, respectively [76,77,78] (Table 3).

#### Clinical Studies

Several clinical studies are ongoing to evaluate the efficacy of miRNAs and EVs as therapeutic agents, although definitive results are currently not available [58,125,161].

One of the ongoing trials investigates the efficacy of EVs isolated from dendritic cells to enhance immunotherapy, while another study evaluates the potential use of EVs for the delivery of chemotherapy drugs [162,163,164] (Table 4).

Taken together, these studies demonstrated the potential benefit and safety of using EVs in the clinical setting [63].

A phase I interventional study which involves the recruitment of patients with malignant glioma to evaluate the efficacy and adverse effects of exosomes that deliver an antisense oligonucleotide is currently underway (https://clinicaltrials.gov: NCT02507583). The study aims to activate the immune system through the exosome release from glioblastoma cells treated with an insulin-like growth factor receptor 1-oligodeoxynucleotide and diffused by implants in the rectum of patients. The researchers hypothesize that immune system activation in glioma patients can reduce the risks and increase the benefits compared to conventional therapy. In another ongoing study, mesenchymal exosomes extracted by stromal cells that incorporate a small siRNA directed against mutated KRAS (https://clinicaltrials.gov: NCT03608631) are used in patients with metastatic pancreatic carcinoma. This clinical trial, like many others, demonstrates the potential use of EVs as a vehicle for small RNAs capable of modulating gene expression and regulate the growth and proliferation of different tumors.

Although the results that emerged from the preclinical studies are encouraging, there are still many limits to overcome before a therapeutic use of miRNAs in clinical practice.

EVs represent a naturally derived system with low immunogenicity, poor toxicity, high stability, which may be modified to improve targeting and loading efficiency [162,165].

On the other hand, some problems limit the use of EVs as potential therapeutic tools: (I) to identify the optimal EVs cellular source for clinical translation; (II) to optimize EVs purification methods (III) to define the techniques suitable for EVs characterization; (IV) to set a regulatory framework for testing EVs as therapeutics in advanced therapy [143].

Other issues do not allow the clinical application of EVs in miRNA delivery [166]. For instance, it is fundamental to optimize methods to load miRNA/antagomiRs into EVs and adequately define the yield of the encapsulated RNA [43,143,167].

To this scope, it is important to understand the molecular mechanisms of miRNA sorting and secretion. Isolated EVs were loaded with miRNAs/anti-miRNAs and delivered either in circulation or into the tumor [119].

Local administration into primary tumors allows reducing off-target effects and adverse effects. Moreover, some preclinical studies demonstrated that systemic administration of exogenous EVs activated macrophages of the mononuclear phagocyte system [28,168].

On these bases, to understand biodistribution and pharmacokinetic profiles of EVs injected by different modalities and routes (acute vs. repeated administration) is needed to reach clinical translation [28,169,170].

Interestingly, several studies have demonstrated that EVs uptake is cell type-specific and EVs can be modified to improve cellular tropism [159,171].

Moreover, optimization of EV-mediated miRNA delivery requires dose-escalation studies of administered EVs encapsulating miRNAs and the definition of the therapeutic window and the maximal dose to be used without saturating the miRNA processing machinery in normal cells [119].

## 4. Conclusions

Data about the role of EV-miRNAs in functional studies are accumulating and their evaluation in the clinical setting is moving its first steps.

However, incomplete knowledge about molecular mechanisms underlying EVs biogenesis and a lack of standardized methods limit the translational potential of EV-miRNAs as diagnostic and prognostic markers. Indeed, we do not know in sufficient detail the biological mechanisms determining the selective formation of EVs cargo, ignoring also how these mechanisms are altered during carcinogenesis and cancer progression.

This aspect restrains our capacity to assign an unbiased diagnostic value to the oncogenic release of EVs. On the other hand, the identification of common EV-miRNA signature has been revealed as a difficult effort, showing high heterogeneity among different studies, mainly due to the numerous differences in experimental methods and the absence of standards. Although the repeatability is currently limited to a few single miRNAs transported by EVs, it is plausible that technical improvements will ameliorate the future diagnostic potential of EV-miRNAs. The new generation RNA analysis methods and the parallel utilization of different models, in vitro, and in vivo, are strategies that can strengthen results in this field.

Moreover, EVs may represent a safe and efficient strategy to deliver miRNAs. Preclinical studies have proven different approaches to increase the therapeutic efficacy of EVs, such as loading methods for specific miRNAs/anti-miRNAs and engineering EVs to express tumor-specific antigen to their surface and improve active targeting of tumor cells. Another promising approach could be based on the tumor microenvironment’s intrinsic capacity to stimulate EVs shedding, opening the way to a strategy based upon adoptive cell transfer, and aimed at perturbing homeostasis in the tumor microenvironment. However, the potential application of EVs in the clinical setting as a cancer therapeutic strategy also requires a better knowledge of EVs biogenesis and functions.

## Figures and Tables

**Table 1 ijms-21-06486-t001:** Pros and cons of the main exosome isolation and miRNA detection methods.

Experimental Procedures	Methodology	Advantages	Disadvantages
**Exosome isolation**	Ultracentrifugation	Bulk exosome purification is easy	Time-consuming, contaminating proteins
	Density gradient centrifugation	High purity exosome	Loss of exosomes, relies on user skill
	Ultrafiltration	Good exosome yield and quick isolation	Less purity
	Immunoaffinity methods	Use of exosome standard markers	Biological properties could be altered due to alterations in markers
	Polymer-based precipitation	Simple and easy procedure	Contamination and retention of polymer
**miRNA detection**	qRT-PCR	Multiplexed, quantitative, high sensitivity and specificity	Sensitive to contaminants, moderately labor-intensive, limited profiling
	Microarray	Comprehensive profiling, Multiplexed	Low specificity and sensitivity, expensive, relative quantitation
	Nanostring	Multiplexed, quantitative, high sensitivity and specificity	The emerging procedure, expensive, limited profiling
	Next-generation sequencing	Comprehensive profiling, Multiplexed, quantitative, high sensitivity, and specificity	Highly labor-intensive, expensive, requires bioinformatic analysis

**Table 2 ijms-21-06486-t002:** EV-associated miRNAs as biomarkers for cancer diagnosis and prognosis.

Tumor	miRNA	EVs Source	Biomarker Potential	Ref
Bladder cancer	miR-125b, -145, -183, and -221	Urine	Diagnostic	[102]
Breast cancer (locally advanced)	miR-503	Serum	Predictive	[76]
(ER+, HER2−)	miR-20b-5p and miR106a-5p	Serum	Diagnostic	[82]
(IDC)	miR-223-3p	Serum	Diagnostic	[86]
(locally advanced)	miR-222	Serum	Prognostic	[89]
(TNBC)	miR-373	Serum	Diagnostic	[92]
Colon cancer	miR-15b, miR-21, miR-25, miR-92, miR-93, miR-223 and miR-486	Serum	Diagnostic	[73]
	Let-7a, miR-1229, miR- 1246, miR-150, miR-21, miR-223, and miR-23a	Serum	Diagnostic	[79]
	miR-150-5p, miR-1246, miR-766-3p, and miR-10b-5p	Serum	Diagnostic	[44]
	miR-125a-3p and miR-320c	Serum	Diagnostic	[81]
Glioblastoma	miR-320 and miR574-3p	Serum	Diagnostic	[83]
	miR-301a	Serum	Diagnostic Prognostic	[90]
NSCLC	let-7b-5p, let-7e-5p, miR-23a-3p, and miR-486-5p	Serum	Diagnostic	[80]
	miR-451a	Serum	Prognostic	[85]
	miR-181-5p, miR-30a-3p, miR-30e-3p, and miR-361-5p	Serum	Diagnostic	[80]
	miR-10b-5p, miR-15b-5p, and miR-320b	Serum	Diagnostic	[80]
	miR-4257 miR-21	Serum	Prognostic	[94]
	miRNA-1-3p, miRNA-144-5p and miRNA-150-5p	Pleural lavage	Diagnostic	[112]
Melanoma	miR-497-5p let7-g-5p	Serum	Prognostic	[95]
	miR-211	Serum	Prognostic	[96]
Ovarian cancer	miR-21, miR-141, miR-200a, miR-200b, miR-200c, miR203, miR-205, miR-214	Serum	Diagnostic	[66]
	miR-200b and miR-200c	Serum	Diagnostic	[91]
	miR-181a, miR-1908, miR-21, miR-486 and miR-223	Serum	Prognostic	[97]
Prostate cancer	miR-200c-3p and miR-21-5p	Serum	Diagnostic	[60]
	miR-1290 and miR-375	Serum	Prognostic	[88]
	miR-222-3p*miR-24-3p/miR-30c-5p	Urine	Diagnostic	[108]
	miR-125b-5p*let-7a-5p/miR-151-5p	Urine	Prognostic	[108]
	miR-200b, miR-200c, miR-30a, miR-375, and miR-99a	Seminal fluid	Diagnostic	[111]

NSCLC: non-small cell lung cancer; ER: estrogen receptor; HER2: ErbB2 receptor; IDC: invasive ductal carcinoma; TNBC: triple-negative breast cancer.

**Table 3 ijms-21-06486-t003:** EV-mediated miRNA or anti-miRNA delivery in cancer therapy.

Strategy	Tumor	miRNA	EVs Isolation Method	Ref
miRNA delivery	Hepatocellular carcinoma	miR-122	Exoquick-TC	[155]
	Breast cancer	miR-134	Exoquick	[141]
	Osteosarcoma	miR-143	Differential centrifugation	[154]
	Glioma	miR-146b	Exoquick-TC	[157]
	Breast cancer	Let-7a	Exoquick-TC	[158]
	Breast cancer	Let-7a	Differential centrifugation	[159]
	Hepatocellular carcinoma	miR-125b	Exoquick TC	[160]
Anti-miRNA delivery	Glioblastoma	mir-9	Differential centrifugation, total exosome isolation kit	[77]
	Sarcoma	miR-150	Differential centrifugation	[78]

**Table 4 ijms-21-06486-t004:** Clinical trials investigating extracellular vesicle delivery in cancer therapy.

Tumor	EVs Source	Study	Ref
Metastatic melanoma	Dendritic cells pulsed with antigen peptides	Phase I	[162]
Colon cancer	Ascites	Phase I	[163]
NSCLC	Dendritic cells pulsed with antigen peptides	Phase I	[164]
NSCLC	Dendritic cells	Phase I–II	NCT01159288
Malignant ascites and pleural effusion	Tumor cell-derived EVs loaded with chemotherapeutic drugs	Phase II	NCT01854866
Colon cancer	Plant exosomes conjugated with curcumin	Phase I	NCT01294072

NSCLC: non-small cell lung cancer.

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
