# Peer review of "Extracellular Vesicles: New Endogenous Shuttles for miRNAs in Cancer Diagnosis and Therapy?"

_ijms, 2020, doi:10.3390/ijms21186486_

Round 1

Reviewer 1 Report

This article is scientifically well composed. 

Author Response

We are very pleased that reviewer has appreciated our manuscript and found it to be relevant for publication.

Reviewer 2 Report

Dear authors,

Your article provides a comprehensive review of EC-miRNA in the content of the cancer diagnosis and therapy. The paper is well written and the overall quality of the article is acceptable. However, there are a number of areas that need to be addressed to improve the manuscript:

  • More titles / subtitles would increase the readability:
    • Section 2.1 is general and could be separately, not under the bodily fluids where blood and urine and other fluids are described.
    • Section 2.2 and 2.3 – add subtitles according to the diagnosis to follow the idea flow
  • Section 2.1: The table on pros and cons of isolation and post isolation techniques would be of great benefit, providing the readers precise information, real review with application into the practice. The mentioned techniques are very briefly described. For example, microarray technology (l.137). However, in this article type I would expect more deep information about utilisation of this method for plasma / serum analyses of miRNAs. When talking about isolation, what is the most reliable miRNA quantification method?
  • Mostly the section 2.2 and 2.3 provide results of several studies. However, it is not mentioned how the authors isolated miRNA and which techniques were used for miRNA detection – either NGS or microarray to have the panel of all known miRNA, or some smaller analyses of some panels? As it is known that different isolation / post isolation techniques provide different miRNA spectrum, are this information reliable?
  • Could you mention in section 2.4 more bodily fluids such as lacrimal and nipple fluids? Were miRNAs isolated also from EV or the whole fluids?
  • Table 1: TNBC is subtype of breast cancer – shift it under the breast cancer line. By the way, specific BC subtypes express different miRNA – which BC subtypes are here as “breast cancer”? (Citation 70,76,80,83).
  • Citation 7 - must be the whole list of co-authors be placed in the citation? It is too long.

Author Response

We thank the reviewer for his valuable suggestions; all the suggested in the revised manuscript.

Following the point by point response.

Reviewer

Your article provides a comprehensive review of EC-miRNA in the content of the cancer diagnosis and therapy. The paper is well written and the overall quality of the article is acceptable. However, there are a number of areas that need to be addressed to improve the manuscript:

More titles / subtitles would increase the readability:

Section 2.1 is general and could be separately, not under the bodily fluids where blood and urine and other fluids are described.

Section 2.2 and 2.3 – add subtitles according to the diagnosis to follow the idea flow.

Authors

A complete revision of the section structure has been performed for the first two chapters. Section 2.1 was moved in the first introductory section and is now section 1.3. Several subtitles have been inserted in sections 2.2 and 2.3 to highlight the main topics of each section better.

Reviewer

Section 2.1: The table on pros and cons of isolation and post isolation techniques would be of great benefit, providing the readers precise information, real review with application into the practice. The mentioned techniques are very briefly described. For example, microarray technology (l.137). However, in this article type I would expect more deep information about utilisation of this method for plasma / serum analyses of miRNAs. When talking about isolation, what is the most reliable miRNA quantification method?

Authors

While we appreciate the reviewer’s comments on this matter, a detailed critical discussion on challenges related to methodology is outside the purpose of this review. Further details could be found in several studies cited in the text. At the same time, advice about these aspects is present in MISEV2014 guidelines. Besides, as reported in 147 and 157 lines, we added further information and relative references about techniques. Moreover, we added the table that describes the advantages and limitations of the primary exosome isolation and miRNA detection methods.

Reviewer

Mostly the section 2.2 and 2.3 provide results of several studies. However, it is not mentioned how the authors isolated miRNA and which techniques were used for miRNA detection – either NGS or microarray to have the panel of all known miRNA, or some smaller analyses of some panels? As it is known that different isolation / post isolation techniques provide different miRNA spectrum, are this information reliable?

Authors

The methodology heterogeneity is a significant challenge in the current interpretation of data from studies of EV as biomarkers. The number of methods, also considering the modifications added in each laboratory, is very high, and render their detailed description outside the possibility of our manuscript. Where necessary, we have indicated the methodology aspects of the studies with their pros and cons. However, a more detailed description and discussion of these aspects would have made less readable the manuscript and compromised the primary purpose to illustrate the current achievements in the EV-miRNA research field. The reader interested in these aspects could consult different studies cited in the text (7, 12, 13, 25, 38, 43, 49, 53, 56, 57, 62, etc.). An explanatory sentence has been inserted at line 129.

Reviewer

Could you mention in section 2.4 more bodily fluids such as lacrimal and nipple fluids? Were miRNAs isolated also from EV or the whole fluids?

Authors

Thank you for this suggestion; lacrimal and nipple fluids were added in section 2.4.

Reviewer

Table 1: TNBC is a subtype of breast cancer – shift it under the breast cancer line. By the way, specific BC subtypes express different miRNA – which BC subtypes are here as “breast cancer”? (Citation 70,76,80,83).

Authors

As correctly suggested by the reviewer, we have shifted TNBC under the breast cancer line and specified the different BC subtypes in table 2.

Reviewer

Citation 7 - must be the whole list of co-authors be placed in the citation? It is too long.

Authors

The reference 7 was added in according to the “Instruction for the authors” guideline. However, we will report this specific case to the editorial office for an eventually changing.

Reviewer 3 Report

Concerning manuscript ijms-903594 by Martelluchi et al. “Extracellular Vesicles: New Endogenous Shuttles for miRNAs in Cancer Diagnosis and Therapy?”.

This review is a nice contribution to our knowledge concerning the basic biological and pathological characteristics of extracellular vesicles in cancer and their possible use in anticancer diagnosis, prognosis and therapy

It is extremely well written and presented in a logical manner that helps our understanding of the field.

There are only a very few places where the syntax or organization of the sentences are not perfect.

One example: on page 11 section 3.1 which is the intrduction to a whole aprt of the manuscript, the information is organized into a series of paragraphs consisting of only one sentence. This makes it harder to follow logically and should be carefully rewritten. Indeed, in the following section 3.1.1, the authors did an excellent job of presentation.

They did the same in some other sections such as the start of section 3.2 on page 12.

I would ask that the authors make these small changes to better the flow of the article.

Further, only as a suggestion- it would helpful if the authors could design a nice figure showing the various aspect discussed and presented

Author Response

Reviewer 3

Concerning manuscript ijms-903594 by Martelluchi et al. “Extracellular Vesicles: New Endogenous Shuttles for miRNAs in Cancer Diagnosis and Therapy?”. This review is a nice contribution to our knowledge concerning the basic biological and pathological characteristics of extracellular vesicles in cancer and their possible use in anticancer diagnosis, prognosis and therapy. It is extremely well written and presented in a logical manner that helps our understanding of the field.

There are only a very few places where the syntax or organization of the sentences are not perfect.

One example: on page 11 section 3.1 which is the intrduction to a whole aprt of the manuscript, the information is organized into a series of paragraphs consisting of only one sentence. This makes it harder to follow logically and should be carefully rewritten. Indeed, in the following section 3.1.1, the authors did an excellent job of presentation.

They did the same in some other sections such as the start of section 3.2 on page 12.

I would ask that the authors make these small changes to better the flow of the article.

Authors

As suggested by the reviewer, we modified the section 3.1 and 3.2 and performed small changes throughout the text to improve the flow of the article and make it easier to read and follow.